# The Removal of Phosphate from Aqueous Solutions by Sepiolite/ZrO₂ Composites: Adsorption Behavior and Mechanism



Željka Milovanović [1], Slavica Lazarević [2,*], Ivona Janković-Častvan [2], Željko Radovanović [3], Slobodan Cvetković [1], Đorđe Janaćković [2] and Rada Petrović [2]

1   Institute of Chemistry, Technology and Metallurgy, National Institute of the Republic of Serbia, Department of Ecology and Technoeconomics, University of Belgrade, Njegoševa 12, 11001 Belgrade, Serbia; zeljka.milovanovic@ihtm.bg.ac.rs (Ž.M.); slobodan.cvetkovic@ihtm.bg.ac.rs (S.C.)
2   Faculty of Technology and Metallurgy, University of Belgrade, Karnegijeva 4, 11000 Belgrade, Serbia; icastvan@tmf.bg.ac.rs (I.J.-Č.); nht@tmf.bg.ac.rs (Đ.J.); radaab@tmf.bg.ac.rs (R.P.)
3   Innovation Center of the Faculty of Technology and Metallurgy, University of Belgrade, Karnegijeva 4, 11000 Belgrade, Serbia; zradovanovic@tmf.bg.ac.rs
*   Correspondence: slazarevic@tmf.bg.ac.rs

**Abstract:** The sepiolite/ZrO₂ composites were prepared by sepiolite (Sep) modification with zirconium propoxide in toluene at room temperature for 24 h (Sep–ZrI) or 95 °C for 4 h (sample Sep–ZrII). The efficiency of the obtained composites for the removal of phosphate from aqueous solutions at initial pH = 4 and pH = 8 was investigated. Characterization of the samples shows that synthesis at a higher temperature for a shorter time provides a slightly higher content of amorphous Zr phase, which is deposited on the sepiolite fibers as a thin layer and agglomerated nanoparticles. Compared to Sep, the composites have a lower point of zero charge and higher specific surface area and pore volume. The adsorption kinetics follow the pseudo second-order model. The adsorption capacities of the composites are approximately the same at both initial pH and higher at initial pH = 4 than at pH = 8. The XPS and ATR-FTIR of Sep–ZrI before and after adsorption identifies the formation of inner-sphere complexes as the mechanism of phosphate adsorption. The slow release during desorption with NaOH solution confirms the strong bonds of the phosphates with the surface of the composites.

**Keywords:** sepiolite; Zr propoxide functionalization; phosphate adsorption; adsorption kinetics; XPS analysis

## 1. Introduction

High concentrations of phosphorus (P) in aquatic environments due to excessive use of fertilizers, discharging of urban and industrial wastewaters, etc., cause the process of eutrophication [1]. During the eutrophication, algae blooms can occur, which leads, over time, to a high number of dead organisms, which are decomposed by saprophytes, where they use oxygen [2]. Consequently, the oxygen concentration in the water decreases and, thus, there is suffocation and mass death of aquatic organisms, which need oxygen for the process of respiration [1,3–5]. In this regard, methods that can effectively remove P from water are being investigated [5].

The two most commonly used methods for the removal of excess P from water are biological processes and chemical precipitation [2,5]. However, these methods have several drawbacks: relatively low efficiency, waste sludge production, and high operating costs [5,6]. On the other hand, it was proven that adsorption is efficient and one of the most economical methods for removing P from water [1,6–8], but only if low-cost adsorbents with a high adsorption capacity are used.

So far, various low-cost natural or waste materials have been tested as adsorbents for P, but their adsorption capacities are very low due to low specific surface area, low affinity of functional groups, and/or negative surface charge, bearing in mind that P is present in the aquatic environment as phosphate anions ($PO_4^{3-}$, $HPO_4^{2-}$, or $H_2PO_4^-$, depending on pH) [9]. Owing to specific adsorption by Lewis's acid–base interactions, many metal oxides/hydroxides have been investigated as adsorbents for phosphates [10]. Due to the remarkable selectivity of phosphate ions and high chemical stability, zirconium oxide/hydroxide has great potential to remove phosphate from water [11,12]. In order to prevent agglomeration and provide a high content of active species for phosphate adsorption, zirconium oxide/hydroxide nanoparticles have been dispersed on different low-cost supports with high surface areas and appropriate porosity, such as zeolite [13,14], bentonite [15–17], mesoporous $SiO_2$ [18], polymeric anion exchanger [19,20], chitosan [21], etc.

To the best of our knowledge, the natural fibrous clay mineral sepiolite has not been used up to now as the support for zirconium oxide/hydroxide particles to prepare the adsorbent for phosphates. Sepiolite ($Si_{12}Mg_8O_{30}(OH)_4(OH_2)_4 \cdot 8H_2O$) belongs to 2:1 trioctahedral silicates, but it is not a "true" layered silicate due to discontinuities of the octahedral sheets and inversion of the silica sheets, whereupon structural tunnels and blocks are formed [22]. The tunnels are filled by zeolitic $H_2O$ molecules and exchangeable cations. Due to such structure, the external surface of sepiolite is rich in silanol groups (Si–OH) and the coordination of octahedral cations is completed by structural $OH_2$ molecules. High specific area, the highest of all clay minerals, is result of fibrous morphology, and the presence of structural channels.

On the other hand, the specific surface area is decreased due to strong hydrogen bonding and Van der Waals' interactions between the fibers, causing formation bundles of fibres. Due to the negative surface charge, sepiolite has a high affinity for cations but a very low affinity for anions.

In general, the deposition of $ZrO_2$ on supports has been carried out in various ways, but usually using $ZrOCl_2 \cdot 8H_2O$ dissolved in water [13,23–26] or $Zr(OC_3H_7)_4$ dissolved in toluene [18,27]. When $ZrOCl_2 \cdot 8H_2O$ was used, $ZrO_2$ was deposited mostly as nanoparticles and $Zr(OC_3H_7)_4$ dissolved in toluene was used to provide a dispersion of $ZrO_2$ functionality at a molecular level, which ensured high surface exposure of adsorption sites and, thus, high adsorption capacity [18,27]. During modification of sepiolite with $Zr(OC_3H_7)_4$, the alkoxy groups of the $Zr(OC_3H_7)_4$ should react with silanol groups on the sepiolite surface, forming Si–O–Zr covalent bonds. By subsequent hydrolysis, the remaining alkoxide groups attached to covalently bound zirconia are converted into a Zr–OH group.

Therefore, in this study, new adsorbents for phosphates were synthesized by adding $Zr(OC_3H_7)_4$ into sepiolite suspension in toluene, under inert atmosphere, and the suspension was mixed: (i) at room temperature for 24 h, similar to the case of silane grafting onto sepiolite [28], or (ii) at 95 °C for 4 h, similar to the case of zirconia functionalization of graphite oxide [27] and SBA-15 [18]. The influence of the synthesis conditions on the properties of Zr–sepiolite and adsorption performances was investigated. The effects of pH, contact time, and initial concentration on phosphate adsorption were examined and the mechanisms of the adsorption discussed.

## 2. Materials and Methods

### 2.1. Materials

The used natural sepiolite (Sep) came from Andrići, (Čačak, Serbia). Characterization of the sample has been performed and reported previously [29].

Zirconium(IV) propoxide ($Zr(OC_3H_7)_4$) (70% in propanol solution) was purchased from Fluka and used as received without further purification. Toluene (Toluene P.A., Lachner) was used for the modification process. All other chemical reagents such as $KNO_3$, NaOH, and HCl were of analytical grade and used as received. Potassium dihydrogen

phosphate anhydrous $KH_2PO_4$ (p.a. Lach-ner) was used to prepare the phosphate stock solution. All solutions were prepared with high purity water (18 MΩ/cm).

### 2.2. Sepiolite Modification with Zirconium(IV) Propoxide

The Sep was modified in a stream of nitrogen, on a magnetic stirrer at a ratio of 10 g of sepiolite and 250 $cm^3$ of toluene, for 30 min, after which 13.5 $cm^3$ of $Zr(OC_3H_7)_4$ was added and left in a stream of nitrogen at room temperature for 24 h in one case (sample Sep–ZrI,) or at 95 °C for 4 h in another case (sample Sep–ZrII). The obtained samples were then centrifuged, and washed with toluene, ethanol, and deionized water. The solids obtained were oven-dried at 110 °C for 24 h.

### 2.3. Characterization of the Samples

The particles morphology of the samples was observed by a Tescan MIRA3 field-emission gun scanning electron microscope (FESEM), with electron energies of 20 kV in a high vacuum. The samples were sputter-coated with an Au alloy to ensure conductivity. Energy-dispersive X-ray spectroscopy of the samples was performed on the device Jeol JSM 5800 Sem (Oxford Link Isis 300, Oxford Instruments, Abingdon, UK). X-ray diffraction (XRD) analysis of the samples was realized with an ITAL STRUCTURES APD 2000 diffractometer using CuKα radiation, in the 2Θ angle range from 3° to 50°, with a 0.02° step. ATR-FTIR spectra of the samples were recorded in absorbance mode using a Nicolet™ iS™ 10 FTIR Spectrometer (Thermo Fisher Scientific, Waltham, MA, USA) with Smart iTR™ ATR sampling accessories, within a range of 4000–400 $cm^{-1}$, at a resolution of 4 $cm^{-1}$, and in 20 scan mode. Thermal behavior was determined by simultaneous TG–DTA (Setsys, SETARAM Instrumentation, Caluire, France) up to 1000 °C with a heating rate of 5°/min in air flow, in an $Al_2O_3$ pan.

The specific surface area ($S_{BET}$) and pore size distribution of the samples were determined using nitrogen adsorption–desorption isotherms obtained by a Micrometrics ASAP 2020 instrument. Prior to adsorption measurements, the samples were degassed at 150 °C for 10 h under reduced pressure. The $S_{BET}$ of the samples was calculated from the linear part of the nitrogen adsorption isotherm according to the method BET [30]. The volume of the mesopores ($V_{meso}$) and pore size distribution were analyzed according to the Barrett, Joyner, and Halenda BJH method [31], using the desorption isotherm. The volume of the micropores ($V_{micro}$) was calculated according to the α-plot analysis [32].

The point of zero charge ($pH_{pzc}$) was determined by the batch equilibration method [33], in $KNO_3$ solutions with a concentration of 0.1, 0.01, or 0.001 mol/$dm^3$. The initial pH values ($pH_i$) of $KNO_3$ solutions (20 $cm^3$) were adjusted by adding of 1 mol/$dm^3$ $HNO_3$ or 0.1 mol/$dm^3$ KOH, in the pH range from 3 to 10. Then 0.02 g of the sample was added to the solution and kept under constant stirring for 24 h at 25 °C. Finally, the samples were filtered and the pH values of the filtrates ($pH_f$) were measured. The $pH_{pzc}$ was obtained from the dependence of $pH_f$ on $pH_i$ as the pH where a plateau appears on the curve $pH_f$ vs. $pH_i$ [29].

XPS analysis of the samples Sep–ZrI before and after adsorption was carried out on SPECS Systems with XP50M X-ray source for Focus 500 and PHOIBOS 100 energy analyzer using a monochromatic Al Kα X-ray source (1486.74 eV) at 12.5 kV and 12 mA. The sample was fixed onto an adhesive copper foil to provide strong mechanical attachment and good electrical contact. Survey XPS spectrum (0–1000 eV BE) was recorded with a constant pass energy of 40 eV, energy step of 0.5 eV, and the dwell time of 0.2 s, while high resolution XPS spectra of the corresponding lines were taken with a pass energy of 20 eV, energy step of 0.1 eV, and a dwell time of 2 s. The XPS spectra were collected by SpecsLab data analysis software, and analyzed using the CasaXPS software package. A standard Shirley background was used for all sample spectra.

### 2.4. Adsorption Experiments

The adsorption experiments were carried out in a batch procedure in a thermostatic water bath with shaking at a temperature of $25 \pm 1$ °C. After separation of the adsorbent, the concentration of phosphate ions was determined by UV/vis spectroscopy Lambda 25/35/45 Perkin Elmer, at 880 nm.

The effect of $pH_i$ on phosphate adsorption was investigated by adjusting a phosphate solution with an initial P concentration of 20 mg/dm$^3$ to different pH values from 3 to 10 and shaking 0.02 g of the Sep–ZrI or Sep–ZrII in 20 cm$^3$ of phosphate solution for 24 h.

For the purpose of determining the adsorption isotherms, solutions of different P concentrations were prepared. An aliquot of 20 cm$^3$ of each solution was shaken for 24 h with 0.02 g of the sample. The adsorption isotherms were determined at $pH_i$ of $4.0 \pm 0.1$ and $8.0 \pm 0.1$.

Kinetic experiments were performed at two concentrations of 5 and 20 mg P/dm$^3$, at pHi $4.0 \pm 0.1$ and $8.0 \pm 0.1$, for contact times ranging from 1 h to 24 h.

The adsorbed quantities of phosphates per unit mass of adsorbents ($q$) were calculated using Equation (1):

$$q = \frac{c_i - c_f}{m} \cdot V \tag{1}$$

where $q$ is the adsorption capacity (mg/g); $c_i$ is the initial phosphate concentration (mg/dm$^3$); $c_f$ is the final phosphate concentration after solution shaking with the adsorbent (mg/dm$^3$); $m$ is the mass of the adsorbent (g); and $V$ is the solution volume (dm$^3$).

The isotherm data were fitted using Langmuir [34] and Freundlich isotherm models [35]. The kinetic data were fitted using the pseudo-first-order model [36], the pseudo-second-order kinetic model [37], and the intraparticle diffusion model [38].

### 2.5. Desorption

Desorption studies were performed with the samples obtained by the phosphate adsorption from a solution with a concentration of 20 mg P/dm$^3$ at $pH_i$ $4 \pm 0.1$ and $8 \pm 0.1$. Desorption was performed for 1–24 h by stirring 0.02 g of phosphate-loaded Sep–ZrI and Sep–ZrII ($m_d$) with 20 cm$^3$ ($V_d$) of 0.1 mol/dm$^3$ NaOH solution at 298 K. The suspensions were then filtered and the phosphate concentration in the solution, $c_{t,d}$, was determined by UV–vis. The quantities of phosphate desorbed per unit mass of the loaded samples, $q_{t,d}$, were calculated using the following equation:

$$q_{t,d} = \frac{c_{t,d}}{m_d} \cdot V_d \tag{2}$$

Desorption efficiency is given as a percentage of the amount of phosphate desorbed per unit mass of the Sep–ZrI or Sep–ZrII, $q_{t,d}$, to the amount of phosphate adsorbed per unit mass of the adsorbent, $q_e$.

### 3. Results and Discussion

#### 3.1. Characterization of the Samples

EDS analysis of the samples (Table 1) shows the presence of O, Mg, and Si, as the main elements of sepiolite and Zr as a result of the modification. Iron is present as an impurity in the Sep [29]. The at.% of Zr in the Sep–ZrII is slightly higher and the at.% of Mg and Si are somewhat lower, indicating a slightly lower content of sepiolite, i.e., a higher content of ZrO$_2$ in the sample Sep–ZrII. Although the differences are not so big, it can be stated that the modification at a higher temperature (95 °C compared to room temperature) for a shorter time (4 h in comparison to 24 h) provides a slightly higher content of deposited Zr compounds.

**Table 1.** Results of EDS analysis of the Sep–ZrI and Sep–ZrII (in at.%).

| Sample | O | Mg | Si | Fe | Zr |
|---|---|---|---|---|---|
| Sep–ZrI | 74.8 ± 0.96 | 7.75 ± 0.33 | 13.2 ± 0.76 | 0.40 ± 0.06 | 3.83 ± 0.07 |
| Sep–ZrII | 77.5 ± 2.39 | 6.52 ± 0.76 | 11.1 ± 1.13 | 0.37 ± 0.07 | 4.57 ± 0.70 |

Diffractograms of Sep–ZrI and Sep–ZrII (Figure 1) display peaks that are typical for the sepiolite, proving that the basic sepiolite structure is unaltered by the modification procedures [29]. The absence of other peaks in the diffractograms of Sep–ZrI and Sep–ZrII in comparison to Sep indicates the formation of amorphous zirconia compounds. The intensities of the sepiolite peaks in the diffractograms of the Sep–ZrI and Sep–ZrII are slightly lower than for Sep, which can be a consequence of a smaller content of the sepiolite due to the presence of amorphous Zr phase.

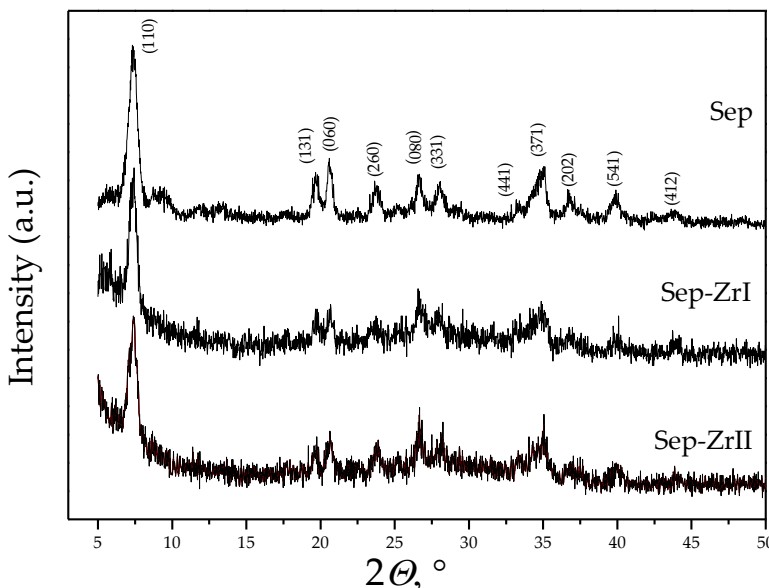

**Figure 1.** XRD patterns of the Sep, Sep–ZrI, and Sep–ZrII.

Differential thermal analysis of the samples (Figure 2) shows peaks characteristic for sepiolite: an endothermic peak at ~100 °C, due to the removal of physically bound water and ethanol used in the synthesis, which is followed by weight loss (TG curve), while an exothermic peak at about 835 °C indicates sepiolite transformation to enstatite ($MgSiO_3$) and $SiO_2$, without the weight loss. The slightly lower intensity of the exothermic peak for Sep–ZrII indicates a slightly lower content of sepiolite in that sample, i.e., a slightly higher content of Zr compounds in comparison to Sep–ZrII, which is in accordance with the EDS results. Weight loss in the temperature range 200–800 °C corresponds to the loss of coordinated and structural (OH group) water, which should be seen as endothermic peaks at DTA curves. However, a wide exothermic peak is seen in that temperature range, which can be explained by the crystallization of the amorphous $ZrO_2$ in the samples [39]. The higher intensity of the peak for Sep–ZrII can be another indication of the higher content of $ZrO_2$ in that sample.

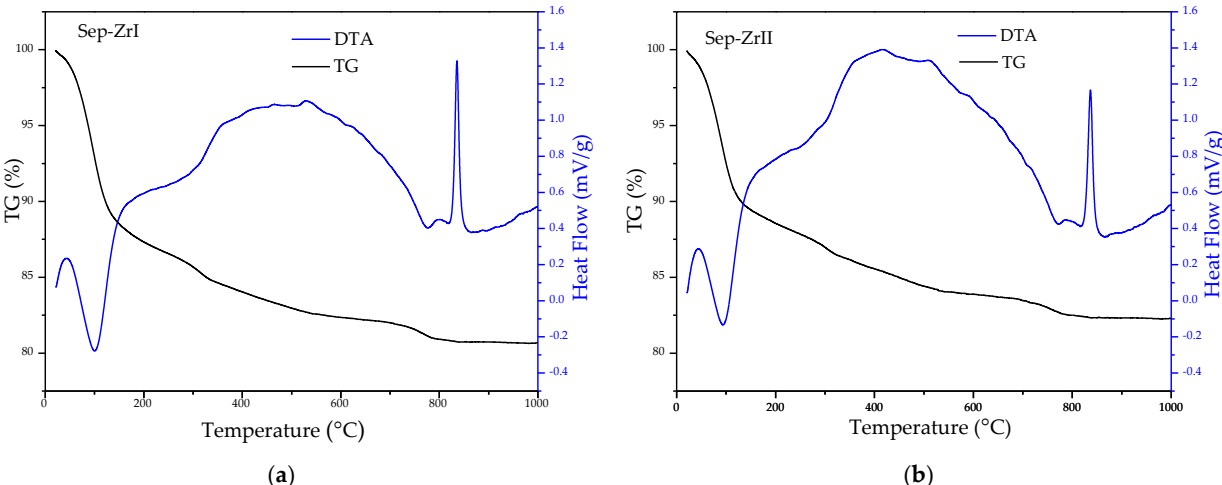

(**a**)   (**b**)

**Figure 2.** DTA and TGA of the samples: (**a**) Sep–ZrI and (**b**) Sep–ZrII.

The SEM micrographs of the Sep–ZrI and Sep–ZrII samples (Figure 3) show the presence of fibrous sepiolite particles and aggregates of fine of subangular particles [40], obviously zirconium oxide/hydroxide. Although the Zr content in the samples is approximately the same, the aggregates of Zr-based particles seem more abundant in Sep–ZrI than in Sep–ZrII (Figure 3). It is possible that the Zr phase is deposited more uniformly as a thin film on the sepiolite particles in the case of Sep–ZrII, which cannot be clearly seen on SEM micrographs. The idea of the modification with $Zr(OC_3H_7)_4$ in toluene is to enable the reaction between silanol groups at the sepiolite surface with the alkoxy group from $Zr(OC_3H_7)_4$ (grafting reaction), not to hydrolyze the alkoxide and precipitate $Zr(OH)_4$ as agglomerated particles. Taking into account the SEM micrographs (Figure 3), it can be supposed that some quantity of alkoxide was not grafted onto sepiolite, but remained adsorbed on the sepiolite surface and hydrolyzed during the samples washing after centrifugation. In that way, agglomerates of fine Zr-based particles were attached to the sepiolite fibers or bundles of fibers, giving amorphous $ZrO_2$ after drying. It is possible that the grafting is more pronounced at higher temperatures, so the quantity of the remaining alkoxide is lower in the case of Sep–ZrII and, thus, the amount of agglomerates.

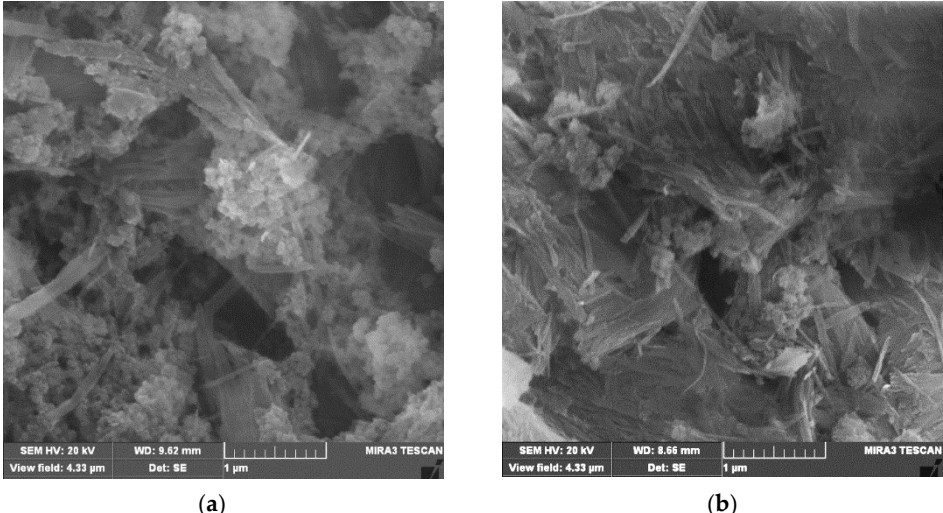

(**a**)   (**b**)

**Figure 3.** SEM micrographs of: (**a**) Sep–ZrI; (**b**) Sep–ZrII.

The calculated $S_{BET}$, $V_{micro}$, $V_{meso}$, the overall pore volume ($V_{total}$), the mesopore size at which the pore size distribution achieves its maximum ($D_{max}$), and the average mesopore diameter ($D_{mean}$) are summarized in Table 2.

**Table 2.** The textural characteristics of the Sep–ZrI and Sep–ZrII in comparison to Sep.

| Sample | $S_{BET}$, m²/g | $V_{total}$, m³/g | $V_{meso}$, cm³/g | $V_{micro}$, cm³/g | $D_{mean}$, nm | $D_{max}$, nm |
|---|---|---|---|---|---|---|
| Sep | 311.4 | 0.351 | 0.265 | 0.126 | 6.63 | 4.00 |
| Sep–ZrI | 337.3 | 0.340 | 0.236 | 0.135 | 6.48 | 4.00 |
| Sep–ZrII | 352.2 | 0.398 | 0.306 | 0.135 | 6.47 | 4.00 |

Specific surface area is increased by the zirconium modification of the Sep. Values of $D_{max}$ and $D_{mean}$ are almost the same for the sample before and samples after modification, which is probably the retention of the basic sepiolite structure. The increase in $S_{BET}$ might be explained by the presence of nanosized zirconium oxide particles on the sepiolite surface (Figure 3). Previous studies show [41] that amorphous $ZrO_2$, which is obviously present in Sep–ZrI and Sep–ZrII, has a large surface area, approximately 380 m²/g. The specific surface area and total pore volume are higher for the Sep–ZrII, i.e., the higher content and probably more homogeneous deposition of zirconium oxide causes the higher surface area. The increase in $S_{BET}$ is also demonstrated in the case of zirconia modification of lingo cellulosic butanol residue [42] and zeolite [43].

The results of the $pH_{pzc}$ determination are shown in Figure 4. It can be noticed from the dependence $pH_f = f(pH_i)$ that the curves for all three concentrations of $KNO_3$ coincide, which indicates that the ions of this electrolyte ($K^+$ and $NO_3^-$) are not specifically adsorbed on the samples, meaning that $KNO_3$ is an inert electrolyte for the Sep–ZrI and Sep–ZrII. According to the pH value of the plateau, the $pH_{pzc}$ of both samples is 7.0 ± 0.1.

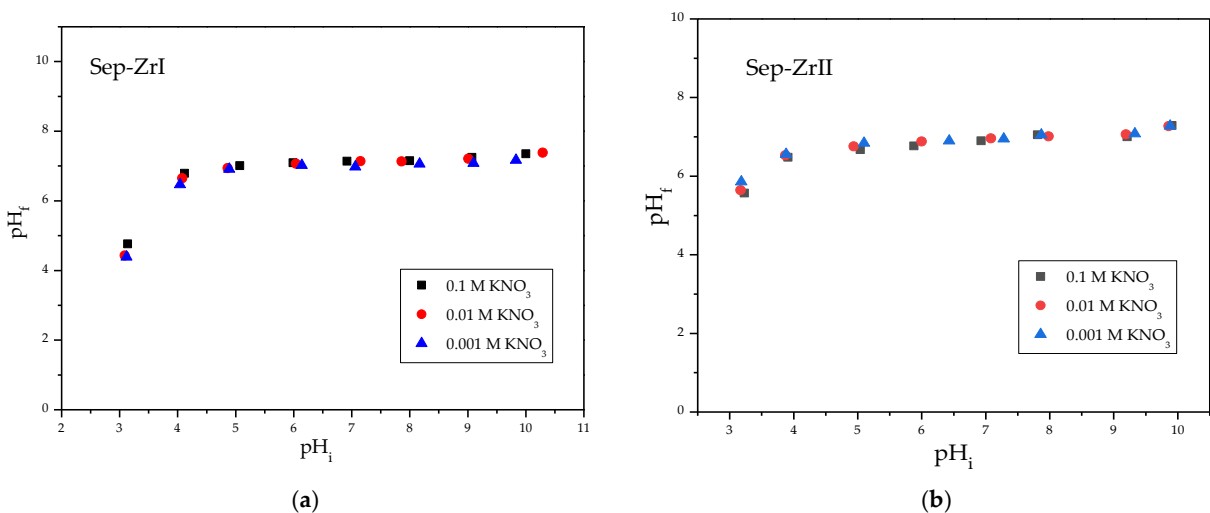

**Figure 4.** $pH_f$ vs. $pH_i$ during the equilibration of (**a**) Sep–ZrI and (**b**) Sep–ZrII with $KNO_3$ solutions.

The $pH_{pzc}$ of natural sepiolite is 7.4 ± 0.1 [29]. Stankovic and authors [39] reported values of 6.6 ± 0.1 and 6.9 ± 0.1 for the $pH_{pzc}$ of the amorphous zirconium oxides, determined in NaCl and $NaNO_3$ solutions. The $pH_{pzc}$ of zirconium (hydr)oxide reported in the literature by other authors were in range 6.7–6.9 [44,45]. The fact that the sepiolite/$ZrO_2$ composites have lower values of $pH_{pzc}$ than the Sep indicates that the surface of the modified sepiolite gain acidity as a consequence of the presence of zirconium oxide, i.e., Zr–OH groups.

### 3.2. Adsorption

3.2.1. Effect of Solution pH

It is found that the adsorption of phosphates is quite dependent on the pH of the starting solutions (Figure 5a) and the highest adsorption capacity is achieved at $pH_i = 3$. When the $pH_i$ increases from 3 to 10, the adsorption capacities of both SEP–ZrI and SEP–ZrII constantly decrease.

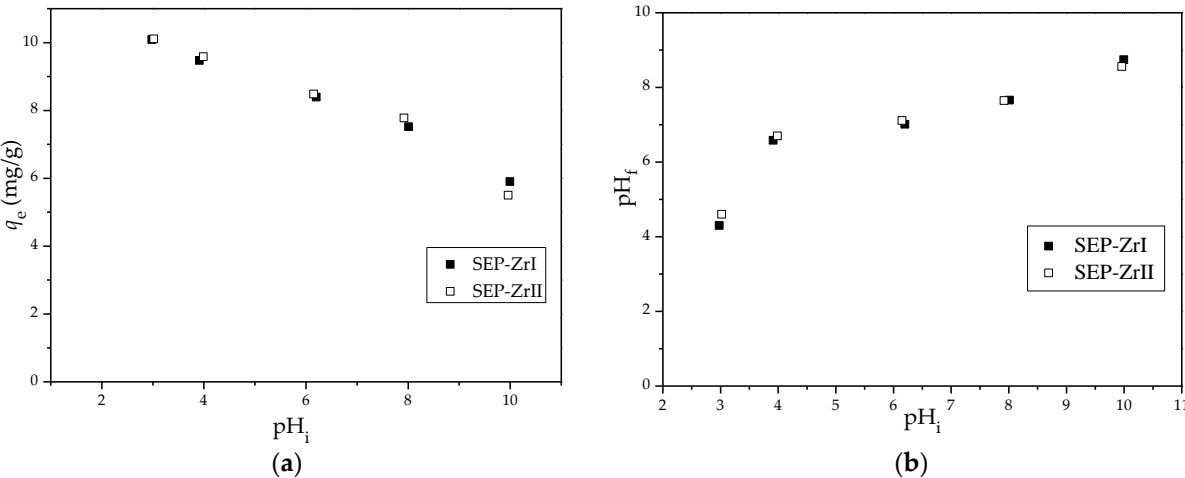

**Figure 5.** (**a**) Effect of $pH_i$ on phosphate adsorption onto Sep–ZrI and Sep–ZrII; (**b**) final solution pH after phosphate adsorption ($pH_f$) onto Sep–ZrI and Sep–ZrII.

The pH of the solution affects the type of phosphate species in aqueous solutions and the charge of functional groups on the adsorbent surface. In the investigated solution pH range (pH 3 to 10), the dominant species of phosphate in the solutions are $H_2PO_4^-$ and $HPO_4^{2-}$. At low pH, the electrostatic attraction between the adsorbent surface and the phosphate anions is enhanced owing to the higher protonation of the adsorbent surface. With an increase in pH value, the proportion of ions with a higher negative charge ($HPO_4^{2-}$) increases, while the positive charge on the surface decreases [7,10], which leads to a decrease in the electrostatic attraction between the phosphate anion and adsorbent and inhibits phosphate adsorption. When solution pH exceeds $pH_{pzc}$ [7,10], the surface becomes negatively charged and the charge increases with increasing pH. Thus, the Coulomb repulsion between the phosphate anion and the negatively charged surface of the modified sepiolite leads to a further decrease in the phosphate adsorption capacity.

At $pH_i < pH_{pzc}$, the equilibrium pH values ($pH_f$) increase (Figure 5b), but are below the $pH_{pzc}$ value. This can be assigned to a protonation of the functional groups on the SEP–ZrI and SEP–ZrII samples (Equation (3)), and the protonated positively charged groups can attract phosphate anions (Equations (4) and (5)):

$$Zr\text{-}OH + H^+ = Zr\text{-}OH_2^+ \tag{3}$$

$$Zr\text{-}OH_2^+ + H_2PO_4^- = (Zr\text{-}OH_2^+)\,(H_2PO_4^-) \tag{4}$$

$$2\,Zr\text{-}OH_2^+ + HPO_4^{2-} = (Zr\text{-}OH_2)_2^{2+}\,(HPO_4^{2-}) \tag{5}$$

At $pH_i > pH_{pzc}$, the $pH_f$ values are lower than $pH_i$, but above the $pH_{pzc}$ value. This can be explained by the release of $H^+$ into solution due to the ionization of the Zr–OH groups and the negatively charged Zr–O⁻ groups are formed. The ratio of the number of Zr–O⁻ and Zr–OH groups increases with the pH incensement. Bearing in mind electrostatic repulsion between phosphate anions and negatively charged surface groups, the adsorption mechanism of the Sep–ZrI and Sep–ZrII at pH > $pH_{pzc}$ can be explained by the exchange

of $H_2PO_4^-$ and $HPO_4^{2-}$ with OH groups from Zr–OH on the adsorbent surface, i.e., by the formation of inter-sphere complexes (Equations (6) and (7)). These complexes are also formed at pH < pH$_{pzc}$.

$$Zr\text{-}OH + H_2PO^{4-} \rightarrow Zr(H_2PO_4) + OH^- \tag{6}$$

$$2Zr\text{-}OH + HPO_4^{2-} = Zr_2(HPO_4) + 2\,OH^- \tag{7}$$

Previous investigations of the mechanism of phosphate adsorption onto $ZrO_2$ show that the replacement of the hydroxyl group bound to zirconium (Zr–OH) with phosphate, i.e., the formation of the Zr–O–P inner-sphere complex is the main mechanism of phosphate adsorption onto $ZrO_2$ at pH > pH$_{pzc}$ [17,46]. The suggested mechanism of phosphate adsorption onto sepiolite/$ZrO_2$ composites is presented in Scheme 1.

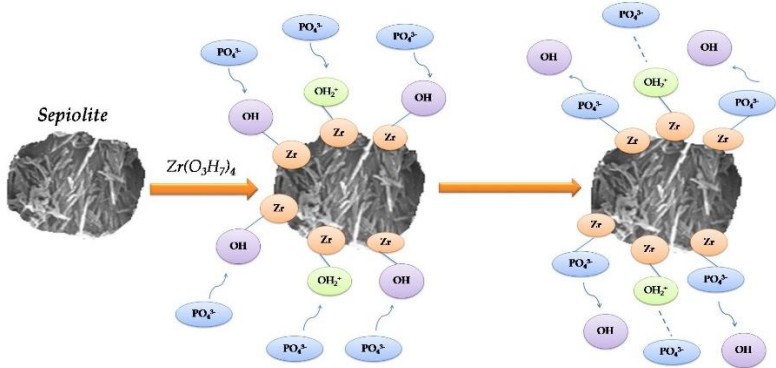

**Scheme 1.** Mechanism of phosphate adsorption onto sepiolite/$ZrO_2$ composites.

### 3.2.2. Adsorption Isotherms

Adsorption isotherms and fitting of the experimental data with the Freundlich and Langmuir nonlinear models are given in Figure 6. The adsorption isotherms constants are summarized in Table 3.

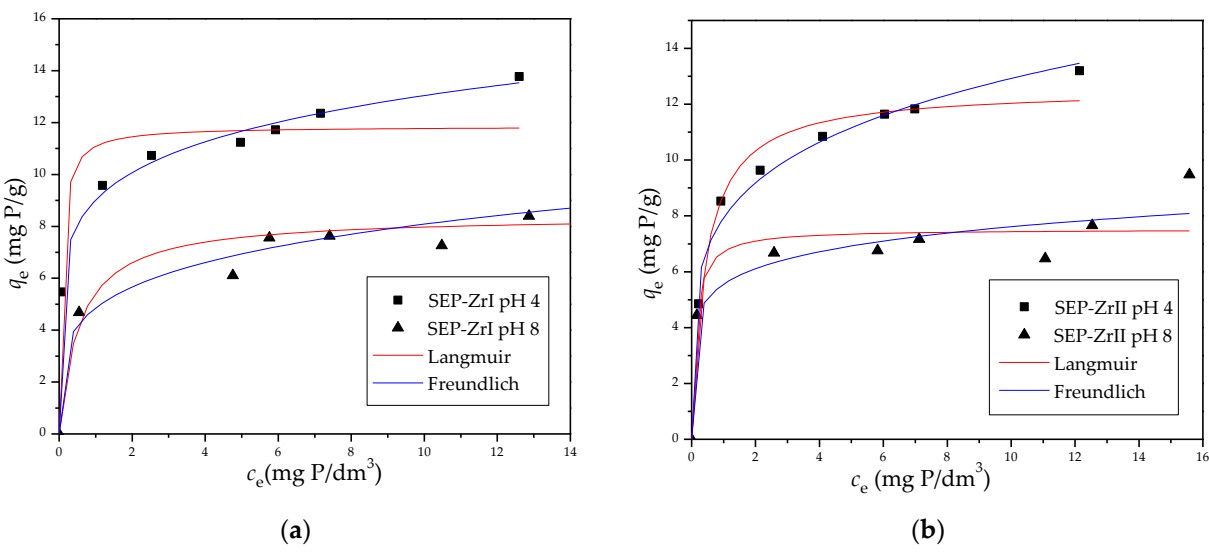

(**a**)　　　　　　　　　(**b**)

**Figure 6.** Adsorption isotherms for phosphate onto Sep–ZrI (**a**) and Sep–ZrII (**b**) at pH of 4 ± 0.1 and 8 ± 0.1.

**Table 3.** Adsorption equilibrium constants obtained from Langmuir and Freundlich isotherms for the adsorption of phosphate onto Sep–ZrI and Sep–ZrII.

| Sample | $pH_i$ | Langmuir Isotherm $q_e = \dfrac{q_m K_L c_e}{1 + K_L c_e}$ | | | Freundlich Isotherm $q_e = K_f \cdot c_e^{1/n}$ | | |
|---|---|---|---|---|---|---|---|
| | | $q_m$ mg/g | $K_L$ dm$^3$/mg | $R^2$ | $1/n$ | $K_f$ (mg/g)(dm$^3$/mg)$^{1/n}$ | $R^2$ |
| Sep–ZrI | 4.0 | 11.85 | 14.45 | 0.935 | 0.160 | 9.01 | 0.996 |
| | 8.0 | 8.47 | 1.81 | 0.905 | 0.221 | 4.85 | 0.957 |
| Sep–ZrII | 4.0 | 12.55 | 2.32 | 0.982 | 0.212 | 7.93 | 0.988 |
| | 8.0 | 7.52 | 8.41 | 0.897 | 0.137 | 5.56 | 0.932 |

Notes: $q_m$ (mg/g)—the maximum adsorption capacity, $K_L$—the Langmuir constant related to the energy of adsorption (dm$^3$/mg), $K_f$—the Freundlich constant related to the adsorption capacity (mg$^{(1-1/n)}$ dm$^{3/n}$/g), $n$—the dimensionless adsorption intensity parameter.

According to the adsorption isotherms, the adsorption capacities of both samples are higher at $pH_i = 4 \pm 0.1$ than at $pH_i = 8 \pm 0.1$, as expected based on the results at Figure 5a. Bearing in mind that the adsorption capacity of sepiolite for phosphates is practically equal to zero, the capacity of Sep–ZrI and Sep–ZrII is the result of the presence of the Zr-phase. The capacity of Sep–ZrI and Sep–ZrII is almost the same at both $pH_i$, regardless of the somewhat larger content (Table 1) and better dispersion (Figure 3) of the Zr phase in Sep–ZrII.

According to the correlation coefficients $R^2$ (Table 3), the Freundlich model describes phosphate adsorption on both samples and both $pH_i$ values better than the Langmuir model. The Freundlich model assumes adsorption on a heterogeneous surface, at energetically non-uniform sites. Thus, phosphate is adsorbed on the functional groups of different energies at the surface of the Zr phase, probably protonated and non-protonated Zr–OH.

The value of the $K_F$ coefficient indicates the adsorption capacity of the adsorbent. An increase in the coefficient in the sequence Sep–ZrI, $pH_i$ $4 \pm 0.1$ > Sep–ZrII, $pH_i$ $4 \pm 0.1$ > Sep–ZrII, $pH_i$ $8 \pm 0.1$ > Sep–ZrI, $pH_i$ $8 \pm 0.1$, indicates an increase in adsorption capacity, which indicates that the highest capacity of the Sep–ZrI sample is at $pH_i = 4 \pm 0.1$, and the lowest capacity of the Sep–ZrI sample is at $pH_i = 8$. Nevertheless, the difference between the samples for the same $pH_i$ is small, indicating a slight difference in adsorption capacity, as already stated.

A comparison of phosphate adsorption by the Sep–Zr samples to other adsorbents containing Zr is presented in Table 4. The phosphate adsorption capacities of the materials used in this work are comparable to the capacity of other similar materials obtained by the deposition of $ZrO_2$ onto different supports.

**Table 4.** The phosphate adsorption capacities of Sep–ZrI and Sep–ZrII in comparison to other adsorbents containing Zr.

| Adsorbent | Capacity | pH | Reference |
|---|---|---|---|
| Zirconia-functionalized graphite oxide | 16.45 mg $PO_4^{3-}$/g | 6 | [27] |
| $ZrO_2$/$Fe_3O_4$ composite | 59.9 mg $PO_4^{3-}$/g | 4 | [47] |
| Amorphous $ZrO_2$ | 99.01 mg $PO_4^{3-}$/g | 6.2 | [11] |
| Zirconium-modified bentonite | 8.90 mg $PO_4^{3-}$/g | 7 | [17] |
| Zirconium(IV)-loaded cross-linked chitosan particles | 71.68 mg $PO_4^{3-}$/g | 3 | [21] |
| La–Zr-modified magnetite | 49.1 mg $PO_4^{3-}$/g | 2 | [48] |
| Magnetic zirconium-based metal–organic frameworks | 12.82 mg P/g | 6.5 | [1] |

**Table 4.** *Cont.*

| Adsorbent | Capacity | pH | Reference |
|---|---|---|---|
| Zirconium-modified zeolite | 10.2 mg P/g | 7 | [14] |
| Zirconium(IV)-loaded lignocellulosic butanol residue | 8.75 mg P/g | 6 | [42] |
| Sep–ZrI | 13.5 mg P/g (41.4 mg $PO_4^{3-}$/g) | 4 | This study |
| Sep–ZrI | 9.8 mg P/g (30.0 mg $PO_4^{3-}$/g) | 8 | This study |
| Sep–ZrII | 13.2 mg P/g (40.45 mg $PO_4^{3-}$/g) | 4 | This study |
| Sep–ZrII | 9.4 mg P/g (28.8 mg $PO_4^{3-}$/g) | 8 | This study |

Compared to amorphous $ZrO_2$, all the presented adsorbents have smaller capacity, which indicates that $ZrO_2$ is active phase for the phosphate adsorption. It should be emphasized that Sep–ZrI and Sep–ZrII have relatively high adsorption capacities at $pH_i = 8 \pm 0.1$, which is the pH of natural waters.

### 3.2.3. Phosphate Adsorption Kinetics

Figures 7 and 8 show the effect of equilibration time on the quantities of adsorbed phosphates on Sep–ZrI and Sep–ZrII, at different $pH_i$ and different initial concentrations.

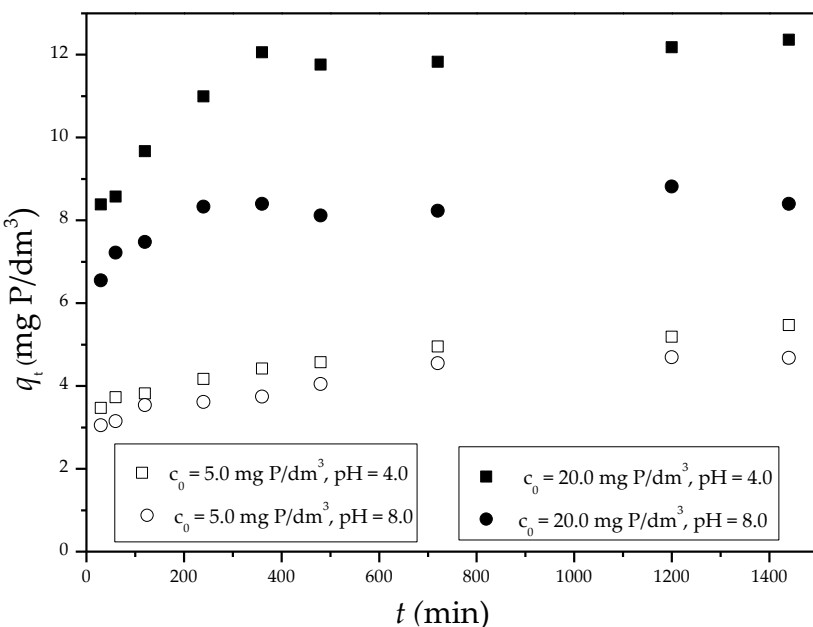

**Figure 7.** Effect of contact time on the adsorbed amount of phosphate onto Sep–ZrI at $pH_i$ 4 $\pm$ 0.1 and 8 $\pm$ 0.1.

Examination of phosphate adsorption kinetics at different pH values and initial concentrations confirms that the adsorption capacity of both samples is higher at $pH_i = 4 \pm 0.1$ than at $pH_i = 8 \pm 0.1$.

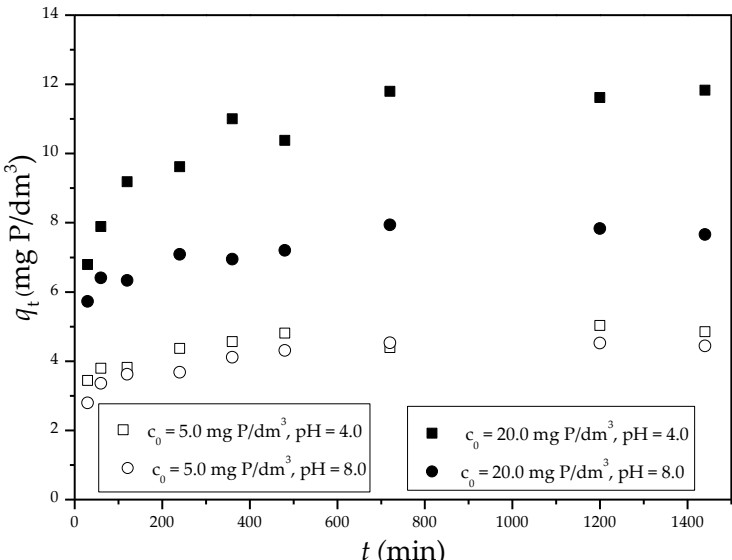

**Figure 8.** Effect of contact time on the adsorbed amount of phosphate onto Sep–ZrII at $pH_i$ $4 \pm 0.1$ and $8 \pm 0.1$.

Phosphate adsorption in all cases takes place in two stages: first, in which the number and availability of sites for adsorption are large, so the driving force for adsorption is also large and the adsorption takes place at a high speed; and second, in which adsorption takes place more slowly because the number of available sites for adsorption is reduced [49]. Table 5 shows the kinetic parameters and correlation coefficients for pseudo-first, pseudo-second-order, and intraparticle kinetic models.

**Table 5.** Kinetic parameters of pseudo-first, pseudo-second-order, and intraparticle model fitting for the adsorption of phosphate on Sep–ZrI and Sep–ZrII, from solutions of different concentrations.

| | Model | | | | | | | | |
|---|---|---|---|---|---|---|---|---|---|
| | **Pseudo-First-Order** | | | **Pseudo-Second-Order** | | | **Intraparticle** | | |
| **Adsorbent/pH** | $\log(q_e - q_t) = \log q_e - \frac{k_1 t}{2303}$ | | | $\frac{t}{q_t} = \frac{1}{k_2 q_e^2} + \frac{t}{q_e}$ | | | $q_t = k_i\, t^{1/2} + C$ | | |
| | $k_1$ (1/min) | $q_e$ (mg/g) | $R^2$ | $k_2$ (g/mg·min) | $q_e$ (mg/g) | $R^2$ | $k_i$ (mg/g·min$^{1/2}$) | $C$ (mg/g) | $R^2$ |
| Sep–ZrI $c_0 = 5.0$ mg P/dm$^3$, pH 4 | 0.0017 | 1.97 | 0.992 | 0.0030 | 5.52 | 0.996 | 0.0598 | 3.22 | 0.988 |
| Sep–ZrI $c_0 = 20.0$ mg P/dm$^3$, pH 4 | 0.0026 | 2.80 | 0.752 | 0.0029 | 12.51 | 0.999 | 0.2842 | 6.60 | 0.989 |
| Sep–ZrI $c_0 = 5.0$ mg P/dm$^3$, pH 8 | 0.0029 | 1.95 | 0.962 | 0.0036 | 4.85 | 1.00 | 0.0648 | 2.68 | 0.950 |
| Sep–ZrI $c_0 = 20.0$ mg P/dm$^3$, pH 8 | 0.0009 | 1.20 | 0.447 | 0.0036 | 8.55 | 0.988 | 0.1368 | 5.99 | 0.939 |
| Sep–ZrII $c_0 = 5.0$ mg P/dm$^3$, pH 4 | 0.0014 | 1.10 | 0.651 | 0.0067 | 4.97 | 0.997 | 0.0807 | 3.05 | 0.975 |
| Sep–ZrII $c_0 = 20.0$ mg P/dm$^3$, pH 4 | 0.0034 | 3.90 | 0.621 | 0.0020 | 12.09 | 0.999 | 0.2858 | 5.54 | 0.952 |
| Sep–ZrII $c_0 = 5.0$ mg P/dm$^3$, pH 8 | 0.0031 | 1.09 | 0.630 | 0.0071 | 4.58 | 1.00 | 0.0743 | 2.63 | 0.930 |
| Sep–ZrII $c_0 = 20.0$ mg P/dm$^3$, pH 8 | 0.0017 | 1.75 | 0.842 | 0.0054 | 7.87 | 0.999 | 0.0880 | 5.46 | 0.910 |

Notes: $k_1$—pseudo-first-order rate constant (1/min), $q_t$—adsorption capacity at time $t$ (mg/g), $q_e$—adsorption capacity at equilibrium (mg/g), $k_2$—pseudo-second-order rate constant (g/mg min), $q_t$—adsorption capacity at time $t$ (mg/g), $q_e$—adsorption capacity at equilibrium (mg/g), $k_i$—intraparticle diffusion rate constant (mg/g min$^{1/2}$) $C$—intercept at the ordinate, related to the boundary layer thickness (mg/g).

Based on the results given in Table 5, it can be seen that the phosphate adsorption process in all cases can be best described by a pseudo-second-order kinetic model, which can indicate that the slowest adsorption step may be chemisorption, which involves the exchange or sharing of electrons between the adsorbent and the adsorbate. The pseudo-second-order model includes all steps of adsorption: external film (boundary layer) diffusion, internal particle diffusion, and adsorption. It is assumed that third step is rapid and, thus, the slowest step would be either film diffusion or pore diffusion.

An intraparticle diffusion model was applied to the results in order to determine which adsorption step was the slowest. In all cases, the dependences $q_t$ vs. $t^{1/2}$ consist of two linear portions (the dependences are not shown) where the first indicates intraparticle diffusion and the second is for equilibrium. The first part does not pass through the origin, which means that intraparticle diffusion is involved in the adsorption, but is not the rate-limiting step.

### 3.2.4. ATR-FTIR Study

The phosphate adsorption mechanism was further investigated by FTIR spectroscopy. Figure 9 shows the FTIR spectra of the Sep, the Sep–ZrI before and after phosphate adsorption, and the Sep–ZrII.

The ATR-FTIR spectra of the Sep–ZrI and Sep–ZrI (Figure 9) are very similar to that of natural sepiolite, indicating that the structure of sepiolite is preserved in the modified samples. Three regions characteristic for sepiolite are observed in Figure 10: (i) bands in the 4000–3000 cm$^{-1}$ range corresponding to the vibrations of the Mg–OH group (3690 cm$^{-1}$), water coordinated to magnesium in the octahedral sheet (3570 cm$^{-1}$), and zeolitic water (at 3422 cm$^{-1}$); (ii) a band at 1658 cm$^{-1}$ due to the vibration of zeolitic water; and (iii) bands in the 1200–400 cm$^{-1}$ range characteristic for silicate structures: bands at 1002 and 460 cm$^{-1}$ due to Si–O–Si vibration; bands at 1205 and 970 cm$^{-1}$ due to Si–O bonds; a band at 437 cm$^{-1}$ originating from octahedral–tetrahedral bonds (Si–O–Mg), and bands at 688 and 642 cm$^{-1}$ corresponding to vibrations of the Mg–OH bond.

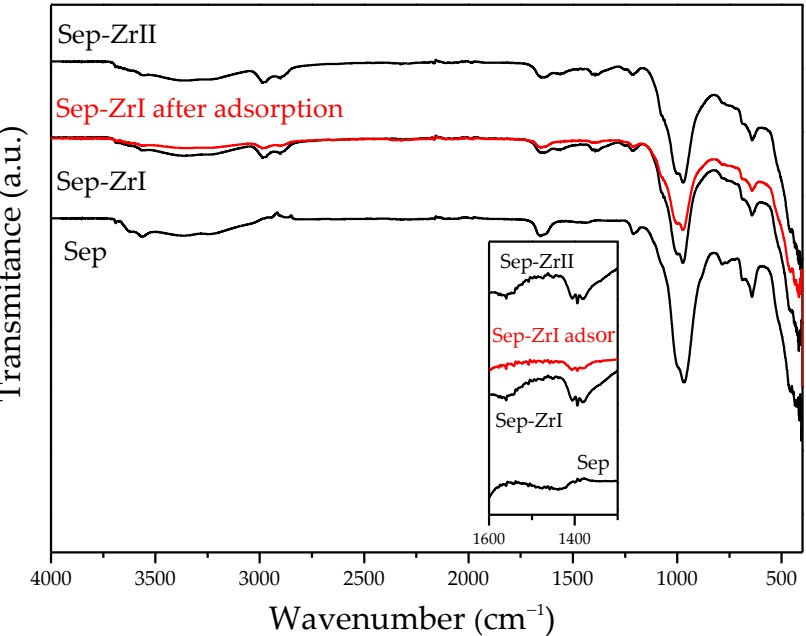

**Figure 9.** ATR-FTIR spectra of samples Sep, Sep–ZrI, Sep–ZrI after phosphate adsorption, and Sep–ZrII.

After modification, weak bands at 1562 cm$^{-1}$ and 1385 cm$^{-1}$ appear, which can be assigned to the vibration of Zr–OH [41]. After phosphate adsorption onto Sep–ZrI, the intensity of these bands significantly decreases, which probably indicates phosphate

adsorption and the replacement of the Zr–OH bond by the Zr–OP bond. The characteristic band that corresponds to the P–O bond at 960 cm$^{-1}$ cannot be clearly seen [41], because it is overlapped with the band for Si–O bonds.

### 3.2.5. XPS Analysis

In order to further clarify the mechanism of the adsorption, XPS analysis of the Sep–ZrI before and after adsorption was conducted and the results are shown in Figure 10.

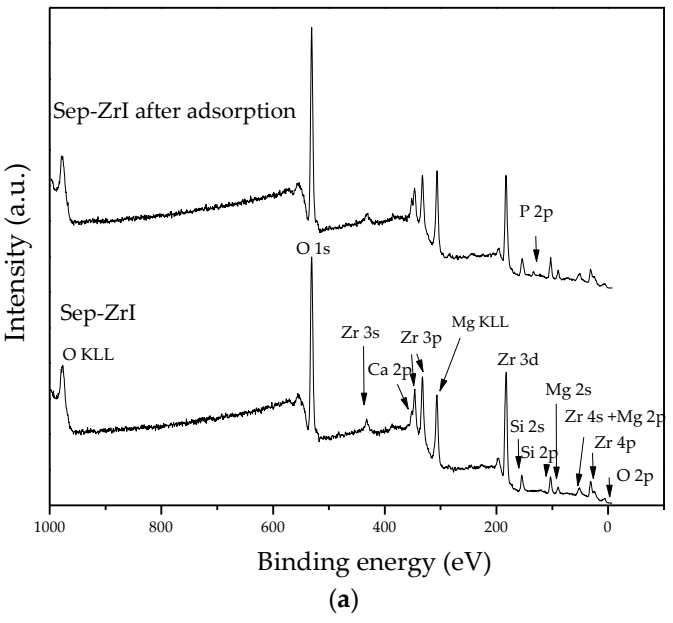

(**a**)

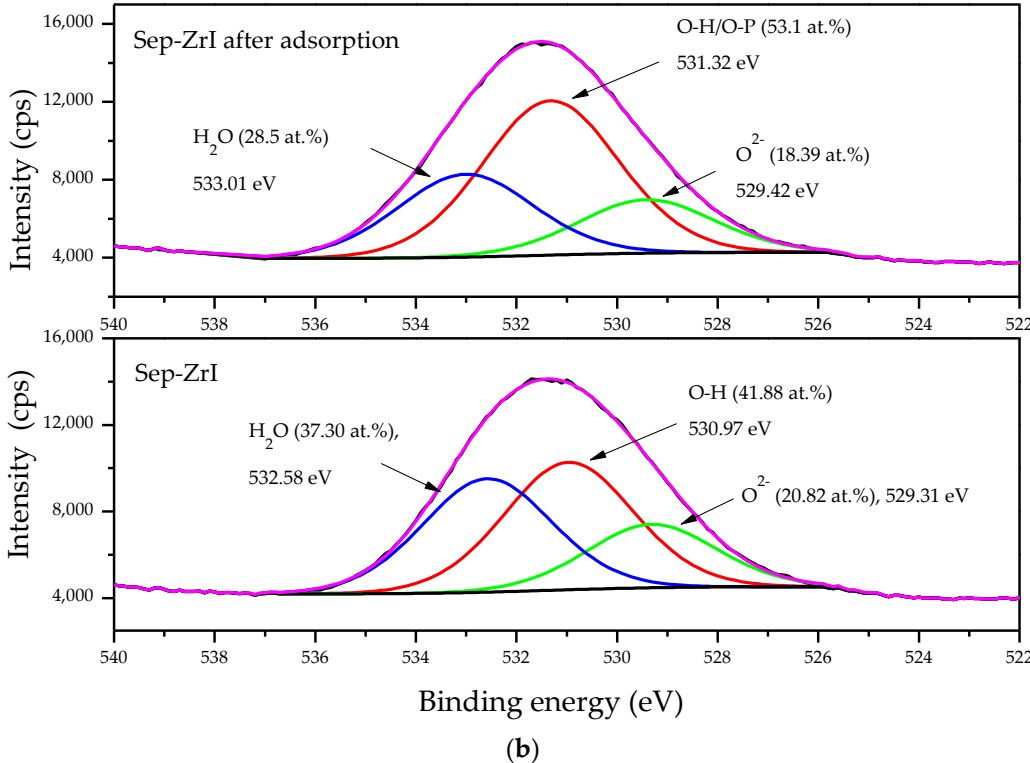

(**b**)

**Figure 10.** XPS spectra of Sep–ZrI (**a**) full-spectrum scanning; and (**b**) O1s orbital spectra of Sep–ZrI before and after adsorption.

The presence of Mg 1s, Mg 2s, Si 2s, Si 2p, Zr 3s, Zr 3p, Zr 3d, and O 1s peaks in the XPS full-scan spectrum of Sep–ZrI (Figure 10a) suggests the presence of sepiolite and Zr phase in the sample. A new peak of P-2p appears in the wide-scan spectrum of the Sep–ZrI after phosphate adsorption, indicating that phosphate has been adsorbed onto the surface of the Sep–ZrI. The P2p binding energy of P-loaded Sep–ZrI is located at 133.6 eV, which is a slightly higher value then the P2p binding energy of $NaH_2PO_4 \cdot 2H_2O$ reference sample [46] (132.0–132.9 eV). This difference can be explained by the formation of strong specific interactions between phosphate and the adsorbent rather than the formation of non-specific electrostatic attraction between the adsorbent and phosphate [46].

The O1s peak of the sample before adsorption can be resolved in three contributions: lattice oxygen ($O^{2-}$), oxygen from surface hydroxyl groups (–OH), and oxygen from adsorbed water ($H_2O$) (Figure 10b). In the case of the sample after adsorption, the peak is also fitted in three contributions, but their fractions are changed: the fraction of oxygen from surface hydroxyl groups increases and the other two proportionally decrease. This incensement can be explained by the replacement of OH groups (from Zr–OH) with $HPO_4^{2-}$ or $H_2PO_4^{-}$, where upon the O–H bond from Zr–OH is replaced by the O–P bond. At the same time, some new O–H groups are included on the surface with $HPO_4^{2-}$ or $H_2PO_4^{-}$. Therefore, the contribution of oxygen from surface OH groups in O1s peak actually corresponds to the oxygen from the O–P and O–H groups. These findings confirm the formation of inner-sphere complexes as the mechanism of phosphate adsorption onto the Sep–ZrI sample. In addition, the positions of all three contributions shift slightly to higher binding energies, illustrating a change in the oxygen environment due to the presence of phosphorus.

### 3.2.6. Desorption

In order to examine the stability and potential regeneration of the sepiolite/$ZrO_2$ loaded by phosphate (phosphate adsorbed at pH 4 ± 0.1 and 8 ± 0.1), desorption was conducted in NaOH solution, concentration of 0.1 M. The dependences of desorption efficiency on contact time are shown in Figure 11.

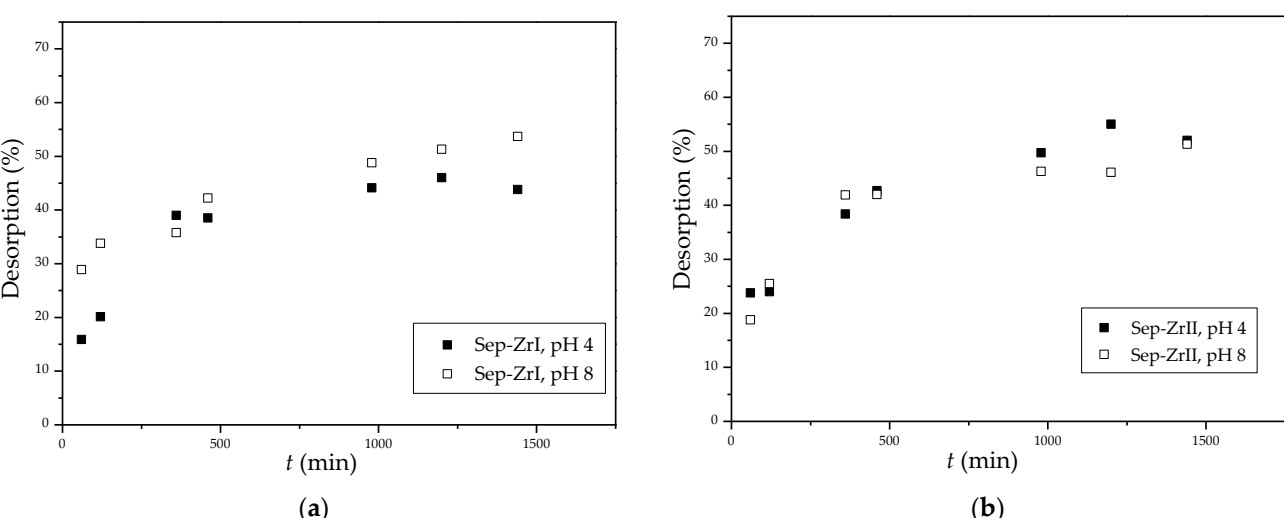

**(a)**                    **(b)**

**Figure 11.** Desorption of phosphate from the saturated Sep–ZrI (**a**) and Sep–ZrII (**b**).

The presented results (Figure 11) indicate slow desorption and the maximum percentage of desorption is approximately 50% after 24 h for all the samples. Obviously, P adsorption on the sepiolite/$ZrO_2$ is not completely reversible. It can be supposed that the easily desorbed phosphates are bonded in the outer-sphere complexes, while the slow-releasing phosphates are in the inner-sphere complexes. Due to the slow release of phosphorus, the zirconium-enriched sepiolite used for phosphate adsorption might be

applied as an artificial fertilizer. Due to the controlled and dosed release of phosphorus, this could be one of the solutions for the soil's phosphorus deficit.

## 4. Conclusions

The sepiolite/amorphous $ZrO_2$ composites were successfully prepared and used for phosphate removal from aqueous solutions. Although zirconium propoxide in toluene was used to provide a molecular level dispersion of $ZrO_2$ functionality, aggregates of nanoparticles were also observed by FESEM in the sample synthesized at a lower temperature. In addition, a higher temperature of synthesis for a shorter time (95 °C, 4 h in comparison to room temperature, 24 h) provides a slightly higher content of $ZrO_2$, but the capacities of both composites at both investigated pH values are practically the same. As expected, capacities are lower at higher pH, but it is important that the composites have a significantly large capacity (~10 mg P/g) at pH close to the pH of natural waters (pH = 8 ± 0.1). The Freundlich isotherm and pseudo-second-order kinetic model fit well to the obtained adsorption data. XPS and ART-FTIR characterization of the sample before and after adsorption indicate the formation of inner-sphere complexes as the mechanism of phosphate uptake by the composites. Desorption studies confirm the formation of inner-sphere complexes, from which the phosphates are released slowly, but also the formation of other-sphere complexes, from which phosphates are easily desorbed.

**Author Contributions:** Conceptualization, R.P. and S.L.; methodology, R.P., S.L. and Ž.M.; formal analysis, Ž.M., I.J.-Č., Ž.R. and S.L.; investigation, Ž.M., I.J.-Č., Ž.R. and S.L.; writing—original draft preparation, Ž.M.; writing—review and editing, R.P. and S.L.; supervision, R.P., Đ.J. and S.C. All authors have read and agreed to the published version of the manuscript.

**Funding:** This work was supported by the Ministry of Science, Technological Development and Innovation of the Republic of Serbia through the project contracts nos. 451-03-47/2023-01/200135, 451-03-47/2023-01/200287, and 451-03-47/2023-01/200026.

**Data Availability Statement:** Data are available upon request.

**Conflicts of Interest:** The authors declare no conflict of interest.

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
