# Peer review of "The Removal of Phosphate from Aqueous Solutions by Sepiolite/ZrO2 Composites: Adsorption Behavior and Mechanism"

_water, doi:10.3390/w15132376_

Round 1
Reviewer 1 Report
Authors have reported the sorption behaviour of Sep-ZrI and Sep-ZrII for phosphate removal. The manuscript may be accepted in “Water” after minor revision according to following remarks.
1. Authors should explain the SEM images by mentioning Fig. 3(a) and Fig. 3 (b). The fine spherical particles were not observed in SEM images as illustrated in the manuscript. Spherical particles should be clearly shown in the images for better understanding of readers.
2. Authors should explain the conduct of isotherm studies at pH4 and 8 though maximum adsorption occurs at pH3.
3. The adsorption parameters such as temperature, adsorbent dose, adsorbate concentration should be studied.
4. The following articles on adsorption may be cited in the introduction section.
(i) Magnetite modified amino group based polymer nanocomposites towards efficient adsorptive detoxification of aqueous Cr (VI): A review, J. Mol. Liq., 337 (2021) 116487, https://doi.org/10.1016/j.molliq.2021.116487.
(ii) Remediation of Cr (VI) using clay minerals, biomasses and industrial wastes as adsorbents, Advanced Materials for Wastewater Treatment, John Wiley Scrivener USA, Advanced Materials Series, 2017, p129-170, ISBN: 9781119407768
(iii) Adsorptive removal of Cr (VI) onto UiO‑66‑NH2 and its determination by radioanalytical techniques, J. Radioanal. Nucl. Chem. 322 (2019) 983-992. DOI 10.1007/s10967-019-06761-w.
5. Please check the sentence “It should be emphasized that Sep-ZrI and Sep-ZrII have relatively high adsorption capacities at pHi = 8 ± 0.1, which is the pH of natural waters. “with the data in Table 4.
6. The mechanism of sorption should be explained with the help of a scheme.
There are some typological and gramatical errors in the manuscript. Authors may rectify these in the revised manuscript.
Author Response
Reviewer 1
Comments and Suggestions for Authors
Authors have reported the sorption behaviour of Sep-ZrI and Sep-ZrII for phosphate removal. The manuscript may be accepted in “Water” after minor revision according to following remarks.
- Authors should explain the SEM images by mentioning Fig. 3(a) and Fig. 3 (b). The fine spherical particles were not observed in SEM images as illustrated in the manuscript. Spherical particles should be clearly shown in the images for better understanding of readers.
Answer: Thank you for your suggestion. In the comment of the SEM micrographs, we wanted to explain that, beside the sepiolite fibers, other particles of different morphology were present in the composites, which are obviously zirconium oxide/hydroxide. According to the suggestion, we changed a description of particle shape, according to the reference (Wang, R.; Ong, E.L.D.; Peerun, I.M.; Jeng, D.S. Influence of Surface Roughness and Particle Characteristics on
Soil–Structure Interactions: A State-of-the-Art Review, Geosci. 2022, 12, 145). The sentence was change to: The SEM micrographs of the Sep-ZrI and Sep-ZrII samples (Figure 3) show the presence of fibrous sepiolite and aggregates of subangular particles.
- Authors should explain the conduct of isotherm studies at pH4 and 8 though maximum adsorption occurs at pH3.
Answer: According to Figure 5a the highest adsorption capacity for both composites was at initial pH value of 3 and decreased with increasing of pH values. Adsorption kinetics was analyzed and adsorption isotherms were determined at initial pH 4 and 8. The initial pH value of 8 was chosen because the pH values of surface waters are around 8 and the adsorbents would be applied to remove phosphate from those waters. The adsorption was not conducted at pH of 3 in order to prevent dissolution of Mg from sepiolite, i.e. changing the sepiolite structure. So, we chose pH values of 4 at which the adsorption capacities were just a slightly lower than for pH of 3.
- The adsorption parameters such as temperature, adsorbent dose, adsorbate concentration should be studied.
Answer: The influence of the adsorbate concentration on the capacity of the adsorbents was shown by adsorption isotherms determination. Adsorbent dose and volume of solution for kinetic investigation and adsorption isotherm determination was chosen according to results of preliminary examinations. The adsorption at temperature of 25 oC was conducted for better understanding of mechanism of adsorption. The influence of temperature on the adsorption capacity will be a part of future research.
- The following articles on adsorption may be cited in the introduction section.
(i) Magnetite modified amino group based polymer nanocomposites towards efficient adsorptive detoxification of aqueous Cr (VI): A review, J. Mol. Liq., 337 (2021) 116487, https://doi.org/10.1016/j.molliq.2021.116487.
(ii) Remediation of Cr (VI) using clay minerals, biomasses and industrial wastes as adsorbents, Advanced Materials for Wastewater Treatment, John Wiley Scrivener USA, Advanced Materials Series, 2017, p129-170, ISBN: 9781119407768
(iii) Adsorptive removal of Cr (VI) onto UiO‑66‑NH2 and its determination by radioanalytical techniques, J. Radioanal. Nucl. Chem. 322 (2019) 983-992. DOI 10.1007/s10967-019-06761-w.
Answer: The suggested references are not related to any aspects of the manuscript. All three references are related to the removal of chromium on different adsorbents, which are neither sepiolite nor ZrO2. Therefore, we could not include the suggested references in the manuscript.
- Please check the sentence “It should be emphasized that Sep-ZrI and Sep-ZrII have relatively high adsorption capacities at pHi = 8 ± 0.1, which is the pH of natural waters. “with the data in Table 4.
Answer: The sentence was clarified in Revised manuscript. The pH values of surface waters are in pH range between 7 and 8. So, the capacity at pH values of 8 was important for the application of the obtained adsorbents for the treatment of the surface waters containing phosphates.
- The mechanism of sorption should be explained with the help of a scheme.
Answer: The scheme explaining the mechanism of adsorption of phosphate onto Sep-ZrO2 composites was added in Revised manuscript.
Reviewer 2 Report
This manuscript was well organized and written. The removal of phosphate from aqueous solutions by sepiolite/ZrO2 composites was studied in details.
One question is that why only two conditions were discussed. It is better to provide more data about sepiolite/ZrO2 composites synthesized at different conditions for optimizing the preparation of sepiolite/ZrO2 composites.
Author Response
Reviewer 2
Comments and Suggestions for Authors
This manuscript was well organized and written. The removal of phosphate from aqueous solutions by sepiolite/ZrO2 composites was studied in details.
One question is that why only two conditions were discussed. It is better to provide more data about sepiolite/ZrO2 composites synthesized at different conditions for optimizing the preparation of sepiolite/ZrO2 composites.
Answer:
Thank you very much for your comments and suggestions.
As we mentioned in the introduction, we used Zr-alkoxide in order to provide dispersion of the ZrO2 functionality at molecular level, where alkoxy-groups of the Zr(OC3H7)4 should react with silanol groups on the sepiolite surface, forming Si–O–Zr covalent bonds. We applied two synthesis conditions, according to the procedures in the literature. In our future work, synthesis conditions will be varied in order to obtain sepiolite/ZrO2 composites with better dispersion, i.e. lower content of ZrO2 but higher adsorption capacities for phosphate.
Reviewer 3 Report
The presented manuscript The removal of phosphate from aqueous solutions by sepiolite/ZrO2 composites: Adsorption behavior and mechanism is devoted to the urgent task of phosphorus removal from waste and natural waters, in particular, using an effective composite adsorbent based on an available and cheap natural mineral - sepiolite.
The article presents exhaustive description of the synthesis and comprehensive characterization of the composite sepiolite/ZrO2 adsorbent, adsorption and desorption behavior of the adsorbent prepared, as well as the mechanism of phosphate adsorption.
The conclusions drawn from the study are consistent with the evidence presented and meet the main purpose of the manuscript.
The scientific data presented in the manuscript are undoubtedly new and original, the experimental evidence given is adequate and exhaustive The manuscript has bees written clearly and logically, and is well structured.
However, I believe that the authors should give the equations describing the phosphate ion adsorption kinetics on the studied adsorbents, the parameters of which are given in Table 5, which would improve the manuscript.
Author Response
Reviewer 3
Comments and Suggestions for Authors
The presented manuscript The removal of phosphate from aqueous solutions by sepiolite/ZrO2 composites: Adsorption behavior and mechanism is devoted to the urgent task of phosphorus removal from waste and natural waters, in particular, using an effective composite adsorbent based on an available and cheap natural mineral - sepiolite.
The article presents exhaustive description of the synthesis and comprehensive characterization of the composite sepiolite/ZrO2 adsorbent, adsorption and desorption behavior of the adsorbent prepared, as well as the mechanism of phosphate adsorption.
The conclusions drawn from the study are consistent with the evidence presented and meet the main purpose of the manuscript.
The scientific data presented in the manuscript are undoubtedly new and original, the experimental evidence given is adequate and exhaustive The manuscript has bees written clearly and logically, and is well structured.
However, I believe that the authors should give the equations describing the phosphate ion adsorption kinetics on the studied adsorbents, the parameters of which are given in Table 5, which would improve the manuscript.
Answer:
Thank you very much for your comments and suggestions. In order to better understand the kinetics results, equations for kinetic models have been added in Table 5.
Reviewer 4 Report
1. Where references follow in the text like [3,4,5,6] consider indicate as [3-6].
2. Section 2.4: Explore recyclability tests to show if the material is reusable.
3. Table 1: Change to at% or mol%, it has more meaning than wt%, since at% indicates the ratio of the atomic species.
4. Figure 1: Label the (hkl) values of XRD peaks.
5. References: Some are not appropriately presented. For example, Reference 10, the correct surname for author number 1 is Liu not Liua, Also Chia and Wanga are Chi and Wang, respectively., the “a” is denoting affiliations as per the reference where it was copied. See how the author list is presented in the publication Ruiting Liu a, Lina Chi a, Xinze Wang a, Yanming Sui a, Yuan Wang b, Hamidreza Arandiyan c . This was also detected in a number of references including 9, 14, 20, etc.
Author Response
Reviewer 4
Comments and Suggestions for Authors
- Where references follow in the text like [3,4,5,6] consider indicate as [3-6].
Answer: Thank you very much for your comments and suggestions. It is done according to suggestion.
- Section 2.4: Explore recyclability tests to show if the material is reusable.
Answer: We performed desorption kinetics and results showed a slow release of adsorbed phosphates. There was no repetition cycle carried out in this case because the goal is to use saturated adsorbents in addition to artificial fertilizers.
Table 1: Change to at% or mol%, it has more meaning than wt%, since at% indicates the ratio of the atomic species.
Answer: wt% was changed to at%.
- Figure 1: Label the (hkl) values of XRD peaks.
Answer: Values of (hkl) of XRD peaks were added.
- References: Some are not appropriately presented. For example, Reference 10, the correct surname for author number 1 is Liu not Liua, Also Chiaand Wangaare Chi and Wang, respectively., the “a” is denoting affiliations as per the reference where it was copied. See how the author list is presented in the publication Ruiting Liu a, Lina Chi a, Xinze Wang a, Yanming Sui a, Yuan Wang b, Hamidreza Arandiyan c . This was also detected in a number of references including 9, 14, 20, etc.
Answer: It is done according to suggestion.